# Personal Care and Household Cleaning Product Use among Pregnant Women and New Mothers during the COVID-19 Pandemic

**DOI:** 10.3390/ijerph19095645

**Published:** 2022-05-06

**Authors:** Andrea L. Deierlein, Alexis R. Grayon, Xiaotong Zhu, Yanwen Sun, Xun Liu, Kaelyn Kohlasch, Cheryl R. Stein

**Affiliations:** 1School of Global Public Health, New York University, New York, NY 10003, USA; xz3255@nyu.edu (X.Z.); ys4613@nyu.edu (Y.S.); xl3593@nyu.edu (X.L.); 2Tandon School of Engineering, New York University, Brooklyn, NY 11201, USA; arg715@nyu.edu; 3Department of Child and Adolescent Psychiatry, Grossman School of Medicine, New York University, New York, NY 10016, USA; kaelyn.kohlasch@nyulangone.org (K.K.); cheryl.stein@nyulangone.org (C.R.S.)

**Keywords:** COVID-19, environmental chemicals, personal care products, household cleaning products

## Abstract

This study examined product use among pregnant women and new mothers in New York City during the COVID-19 pandemic (July 2020–June 2021). Women reported use of personal care and household cleaning products within the previous month, changes in antibacterial product use, receipt of healthcare provider advice, and opinions on environmental chemicals (n = 320). On average, women used 15 personal care products and 7 household cleaning products. Non-Hispanic Black women used nearly two more personal care products; non-Hispanic Black women, those with a college degree, and essential workers used 1–3 more household cleaning products. Women who were Hispanic or reported their race and ethnicity as Other were two times more likely to use antibacterial personal care products. Non-Hispanic Black, Hispanic, and women who reported their race and ethnicity as Other were 1.5 times more likely to increase antibacterial product use during the pandemic. Nearly all women agreed that environmental chemicals pose health risks and are impossible to avoid, while less than one quarter received advice regarding product use. Product use is a modifiable source of chemical exposures. Results from this study suggest that women may have increased their product use during the pandemic. Healthcare providers may use the current focus on health hygiene to promote discussion and assessment of environmental chemical exposures with patients.

## 1. Introduction

Personal care products and household cleaning products are common sources of environmental chemical exposures. These chemicals include phthalates, parabens, cyclosiloxanes, anti-microbial/bacterial agents, bisphenols, phenols, and ultraviolet filters, all of which have suspected endocrine-disrupting or asthma-associated properties [1,2]. Endocrine-disrupting chemicals interfere with normal endocrine hormone function; they can block or mimic natural hormone actions, resulting in a range of adverse health effects throughout the body, especially when exposures occur during early life [3,4,5]. Women often have higher chemical exposure levels than men, which may be attributed to their greater use of personal care products (e.g., hair products, cosmetics, perfumes, lotions). Prenatal exposures to environmental chemicals, assessed by maternal biomarkers or self-reported product use, are associated with numerous reproductive, metabolic, and neurobehavioral health risks in the offspring [3,6,7,8,9,10,11] as well as the mother, spanning preconception through to postpartum [4,12,13,14].

Product use, particularly of personal care products, is prevalent among pregnant women and those of reproductive age. Average daily use ranges from approximately 6–12 personal care products, with some women reporting use of up to 30 products every day [15,16,17,18,19,20,21]. Differences in types and frequency of personal product use by race and ethnic groups exist, which may contribute to health disparities in these populations [2,15,22]. Less is known about household cleaning product use among women; however, use of various cleaning sprays and air fresheners during pregnancy has been documented [9,10]. Recently, the COVID-19 pandemic presented a unique situation that altered usual behaviors related to purchases (and presumably use) of personal care and household products [23,24]. In analyses of consumer-spending survey data from the U.S. and other countries, spending on household supplies increased, while spending on personal care products, particularly cosmetics and fragrances, decreased at the beginning of the pandemic [23,24,25]. There is limited data on purchases/use of specific products (e.g., nail polish, body lotion, antibacterial products) and among specific populations (e.g., women of reproductive age). In the current study, our objective was to examine personal care and household cleaning product use among pregnant women and new mothers in New York City during the first year of the pandemic. We also examined women’s opinions of environmental chemicals and their receipt of healthcare provider advice regarding product use during this time. 

## 2. Materials and Methods

The purpose of the Maternal Health and Behavior Study (MHBS) was to examine lifestyle behaviors and household and personal product use among pregnant women and new mothers during the COVID-19 pandemic. MHBS participants were recruited from the COVID-19 and Perinatal Experiences Study (COPE). COPE is an observational study collecting the demographic, social, occupational, and health-related information of pregnant women and new mothers (defined as women with a live birth within the previous 6 months) during COVID-19. Beginning in May 2020, women with a pregnancy diagnosis (medical code Z34) within the New York University Langone Health System were sent an electronic message inviting them to participate in COPE. Women were ineligible if they were younger than 18 years, were no longer pregnant, did not have a live birth, or were not fluent in English or Spanish. Women completed up to six remote survey assessments spaced approximately 2 months apart. At the end of the second remote assessment, women received an invitation to participate in MHBS, which required the completion of an additional remote survey (~30 min) querying their health behaviors, including physical activity, dietary intakes, and personal care and household cleaning product use. Women were compensated with a $12 gift card for completing the survey. All women provided written, informed consent; COPE and MHBS protocols were approved by the institutional review board of the NYU School of Medicine.

### 2.1. Personal Care and Household Cleaning Product Use, Receipt of Healthcare Provider Advice, and Opinions of Environmental Chemicals

A list of 22 types of personal care products and 17 types of household cleaning products was adapted from the Early Life Exposures Assessment Tool [26] and previous studies [9,16]. Personal care products included soaps, body and face lotions, hair styling products, deodorant, sunscreen, nail polish, and perfume (full list in Appendix A). Household cleaning products included bleach, ammonia, multi-use cleaners (without bleach or ammonia), other cleaning sprays, and air fresheners (full list in Appendix A). Women reported their frequency of use (daily, weekly, monthly, never, or don’t know) of products during the previous month. They reported on how their use of personal care products and household cleaning products labeled as “antibacterial or antimicrobial” and those labeled as “organic, environmentally/eco-friendly, chemical-free, green, or natural” changed from before the COVID-19 pandemic (March 2020) to the current time period at survey administration (more often, about the same, less often, or never). Women were asked whether their prenatal healthcare provider had discussed: (1) personal care product use, such as which types of products contain ingredients that are safer to use and which ingredients should be avoided (yes, somewhat, no) and (2) household cleaning product use, such as which types of household cleaners are safer to use and which types should be avoided (yes, somewhat, no). Lastly, they were asked about their opinions on environmental chemicals by their level of agreement (strongly agree, agree, neither, disagree, strongly disagree) with the following statements: “Chemicals in the environment can pose health risks” and “Chemicals in the environment are in so many things that it’s impossible to avoid them.” These questions were adapted from those used in The Infant Development and the Environment Study (TIDES) [27]. 

### 2.2. Sociodemographic and Other Characteristics

Data were collected on age (to the nearest year); race and ethnicity (categorized as non-Hispanic Black, non-Hispanic Asian, Hispanic, non-Hispanic White, or Other Race, which included Native American or Alaska Native, Native Hawaiian or Pacific Islander, self-report of “Other”, and self-report of two or more races); highest educational attainment (categorized as less than college, college graduate, and professional/graduate school); marital status (dichotomized as married/partnered or single/separated/divorced); and employed as essential worker (“During the COVID-19 outbreak, are you considered an essential worker?”, yes or no). The time period of survey administration was categorized as July–October 2020, November 2020–February 2021, and March–June 2021 to generally reflect three pandemic-related factors: seasonality (changes in weather and temperature to reflect spring/summer/fall versus winter, when concern about COVID transmission may be higher); vaccine availability (vaccines became broadly available beginning in March 2021); and regional infection rates [28].

### 2.3. Statistical Analyses

All statistical analyses were conducted in Stata/SE 15.1 (StataCorp LP, College Station, TX, USA). We performed descriptive analyses of women’s characteristics for the entire sample and stratified by maternal status, currently pregnant or new mother, using *t*-tests (continuous variables) or chi-square tests (categorical variables). We used one-way ANOVA tests to examine differences in the number of types of personal care products and the number of types of household cleaning products used during the previous month (means and standard deviations, SD, of number of types of products used) by the selected characteristics (timing of survey administration, maternal status, age, race and ethnicity, education, and essential worker status). Multivariable linear regression models estimated adjusted mean differences in the number of types of personal care products and household cleaning products used (separate models) with the selected characteristics (models included all characteristics simultaneously). 

Chi-square tests examined bivariate associations of women’s change in use of antibacterial and organic products during the pandemic (more often compared to about the same/less often/never, reference), receipt of healthcare provider advice (yes/somewhat compared to no, reference), and opinions of environmental chemicals (strongly agree/agree compared to neither/disagree/strongly disagree, reference) with the selected characteristics. We used separate multivariable robust Poisson regression models to estimate adjusted prevalence ratios (aPR) for change in product use (antibacterial and organic personal care and household cleaning products, respectively), receipt of healthcare provider advice, and agreement that environmental chemicals are impossible to avoid with the selected characteristics (models included all characteristics simultaneously). We did not examine associations for agreement that environmental chemicals can pose health risks, since 94% of women reported agreement. 

## 3. Results

There were 424 women who completed the second assessment of COPE and were invited to participate in the MHBS; 340 women consented for the MHBS, of which 320 (75%) completed all survey questions. The majority of women were non-Hispanic White (63%), college-educated or higher (87%), and married or living with a partner (95%). Over a third (36%) of women were essential workers and 66% of women completed the survey between July and October 2020 (Table 1).

Women used a mean (SD) of 14.7 (3.5) types of personal care products and 7.4 (3.6) types of household cleaning products during the previous month. The personal care products most commonly used on a daily basis (by at least 50% of women) were hand sanitizer, soaps, lotions, and deodorant (Appendix A). The household cleaning products most commonly used on a daily basis (by at least 10% of women) were multi-use cleaners (defined as “multi-use cleaners that do not contain bleach or ammonia”), bleach, and air fresheners (Appendix A). 

In multivariable linear regression analyses, the adjusted mean number of types of personal care products and/or household cleaning products used varied by women’s characteristics (Table 2). Non-Hispanic Black women used nearly two (adjusted B 1.68, 95% CI: 0.18–3.18) more types of personal care products during the previous month compared to non-Hispanic White women. Greater household product use, ranging from approximately 1–3 more types of products, was observed among non-Hispanic Black women (adjusted B 2.74, 95% CI: 1.24–4.24), women with a college degree (adjusted B 0.92, 95% CI: 0.03–1.81), and essential workers (adjusted B 1.19, 95% CI: 0.34–2.04).

Use of antibacterial products increased during the pandemic (Table 3); 33% and 53% of women reported using antibacterial personal care products and antibacterial household cleaning products, respectively, more often during the pandemic compared to before the pandemic (March 2020). In multivariable robust Poisson regression models (Table 4), women who were Hispanic or reported their race and ethnicity as Other were approximately 2 times more likely to use antibacterial personal care products than non-Hispanic White women (aPR 1.97, 95% CI: 1.22–3.19 and aPR 1.73, 95% CI: 1.11–2.68, respectively). Non-Hispanic Black, Hispanic, and women who reported their race and ethnicity as Other were approximately 1.5 times more likely than non-Hispanic White women to report increasing their use of antibacterial household cleaning products during the pandemic (aPR 1.57, 95% CI: 1.15–2.13; aPR 1.83, 95% CI: 1.35–2.47; aPR 1.64, 95% CI: 1.25–2.15, respectively). Use of organic products also increased during the pandemic (Table 3); 23% and 26% of women reported using organic personal care products and organic household cleaning products, respectively, more often during the pandemic compared to before the pandemic. No differences in organic product use were observed by maternal characteristics (Table 4), with the exception that women aged 35 years and older were less likely to use organic personal care products more often during the pandemic (aPR 0.47, 95% CI: 0.28–0.79).

Only 23% of women received any advice regarding personal care product use from their healthcare providers (13%, n = 41 reported “somewhat” and 10%, n = 33 reported “yes”) and only 16% of women received any advice regarding household cleaning product use from their healthcare providers (9%, n = 29 reported “somewhat” and 7%, n = 23 reported “yes”). Differences were observed by race and ethnicity, with greater percentages of women of color receiving advice compared to non-Hispanic White women (Table 3). Hispanic women were 2 times more likely to receive advice compared to non-Hispanic White women (aPR 1.79, 95% CI: 0.97–3.31 and aPR 2.07, 95% CI: 1.02–4.19 for personal care products and household cleaning products, respectively, Table 4). Nearly all women agreed with the statement that environmental chemicals pose health risks (94% total; 55%, n = 175 reported “strongly agree” and 40%, n = 128 reported “agree”) and that environmental chemicals are impossible to avoid (80% total; 26%, n = 82 reported “strongly agree” and 55%, n = 175 reported “agree”); no differences were observed in women’s level of agreement by the selected characteristics (Table 3 and Table 4).

## 4. Discussion

Among the 320 pregnant women and new moms assessed in New York City during the COVID-19 pandemic, women reported using approximately 15 types of personal care products (out of 22 types) and 7 types of household products (out of 17 types), on average, during the previous month. While frequency of product use did not differ by timing of survey completion, women reported that their use of antibacterial and organic products increased compared to before the pandemic; 33% and 53% of women used antibacterial personal care products and household cleaning products, respectively, more often during the pandemic compared to before the pandemic. Differences were observed by women’s race and ethnicity; compared to non-Hispanic white women, non-Hispanic Black women used more types of personal care and household cleaning products during the previous month. Women who were non-Hispanic Black, Hispanic, or reported their race and ethnicity as Other were also more likely to increase their use of antibacterial products during the pandemic. The overwhelming majority of women viewed environmental chemical exposures as health risks and believed that they were impossible to avoid, yet over three quarters of women reported that their healthcare provider had not discussed product use with them.

Previous studies examined product use among pregnant women [9,10,16,17,18,19] and women of reproductive age [15,20,21,29], with the majority focusing on personal care products [15,16,17,18,19,20,21,29]. Although studies varied by the number and types of personal care products queried, all studies showed that women used a range of chemical-containing products on a daily basis. Toothpaste, deodorant, lotions, soaps, sunscreen, and makeup/cosmetics, including those that were scented, were among the most frequently used products. Fewer studies have considered household cleaning product use. Among cohorts in Spain and England, women reported using disinfectants, bleach, furniture polishes, glass cleaners, and air fresheners during pregnancy [9,10]. The findings of the current study add to this literature and highlight that some women increased their use of antibacterial- and organic-labeled products in response to the COVID-19 pandemic. These products may contain fragrances and other chemicals (including replacement chemicals for triclosan, a banned antibacterial and antifungal agent [30]), despite sometimes being advertised as “natural” or “green” [1]. Greater frequency of antibacterial product use (“more often”) was observed, particularly among women who identified as non-Hispanic Black, Hispanic, or Other race and ethnicity. Racial and ethnic differences in product use (such as types, frequency of use, and duration), as well as endogenous hormonal activity, have been documented and may contribute to disparities in risk of hormonally mediated diseases [15,22,31,32]. For example, greater personal care product use, especially hair products, and higher exposures to chemicals from personal care products are often observed among Black women and are linked to increased risk of breast cancer, and other reproductive health-related conditions, in this population [2,15,22,33,34,35]. 

Product use is associated with chemical body burden in women. Use of lotions, gels, cosmetics, perfumes, hair spray, nail polish, and deodorant is associated with greater urinary concentrations of phthalate, phenol, and paraben biomarkers [17,18,29,36,37]. These chemicals have suspected endocrine-disrupting properties, displaying estrogenic and anti-androgenic capabilities, and are found in a wide range of medical, consumer, industrial, and food products, in addition to personal care products [17,38]. Exposures to endocrine-disrupting chemicals, assessed using biomarkers and/or self-reported product use, are linked to reduced fecundity/fertility, pregnancy complications, and later-life cardiometabolic health outcomes in women [4,12,13,14,39,40]. During pregnancy, chemicals can enter fetal circulation and have been measured in cord blood and meconium in relation to maternal exposure levels [41,42]. Prenatal exposures may influence fetal development and have been associated with reduced gestational size and age [6,43,44], attention deficit disorders and related behavioral outcomes [3], weight status [45], respiratory health outcomes [8,9,10], and altered pubertal timing [46] in the offspring. Given the health risks posed by environmental chemicals for women and their offspring, it is necessary to implement strategies to reduce exposures among this population. 

The importance of discussing and assessing chemical exposures during routine healthcare visits has been acknowledged by obstetricians and other reproductive health specialists [47,48,49]. However, there is limited evidence that clinicians put these actions into practice [50,51]. Clinicians report barriers related to inadequate knowledge and training regarding environmental chemicals; not enough time during visits and concern about causing stress to their patients; and lack of clinical evidence and established guidelines for reducing health risks attributed to environmental chemicals [50,51]. In the current study, only a small percentage of women reported that their healthcare provider had advised them on product use. Other studies suggest that media, internet, friends, and family serve as the main informational sources on chemical exposures from products [15,21]. Given that many women believe that environmental chemicals are dangerous and impossible to avoid [27], healthcare providers should be empowered to counsel women about how to minimize their exposures and understand the potential risks from exposure. Proposed strategies include increased educational opportunities for providers (e.g., medical school curricula and continuing education, publication of environmental health studies in general reproductive health and primary care journals) and dissemination of provider-centric and patient-centric resources, such as those available through the University of California San Francisco Program on Reproductive Health and the Environment (Available online: https://prhe.ucsf.edu accessed 12 March 2022) [50,51]. Additionally, providing women with feasible individual-level strategies, such as discontinuing use of some products and purchasing products labeled as “chemical-free”, may be successful in reducing personal chemical exposures [51,52,53].

A main strength of this study was the evaluation of both personal care products and household cleaning products among a diverse, urban cohort of pregnant women and new mothers during the COVID-19 pandemic. We also queried women’s change in antibacterial product use, opinions on environmental chemicals, and receipt of healthcare provider advice regarding exposures during this time. Our study was limited by our evaluation of products. Women reported on their frequency of use of selected types of products; for example, we only queried “face makeup” rather than a range of face cosmetics, such as makeup primers, concealers, powders, and mascara. We also lacked specific brand information and chemical constituent classes. We did not collect biomarker data; however, questionnaire data that includes a range of personal care and household products is likely useful for assessing chemical exposures during this sensitive time period [17]. Women also completed the questionnaire at different time periods throughout the pandemic. Interestingly, we did not observe meaningful differences in product use in relation to regional variation in infection rates, although two-thirds of women responded during summer and fall months when numbers of COVID cases in New York city were low [28].

## 5. Conclusions

Product use is a potentially modifiable source of environmental chemical exposures among women. We found that pregnant women and new moms used a range of personal care and household cleaning products and increased their use of antibacterial products in response to the pandemic. Differences were observed by women’s race and ethnicity, level of education, and employment as an essential worker. Despite the majority of women having negative perceptions of environmental chemical exposures, only a small percentage received any advice from their healthcare providers. Given the current focus on health hygiene, healthcare providers may be able to take this opportunity to promote discussion and assessment of chemical exposures with their patients.

## Figures and Tables

**Table 1 ijerph-19-05645-t001:** Distributions ^1^ of selected characteristics among women participating in the Maternal Health and Behavior Study (n = 320).

Characteristic	Total		Pregnant (n = 98)		New Mother (n = 222)	
	n	%	n	%	n	%
Time Period of Survey Completion						
July–October 2020	217	66	58	58	159	70
November 2020–February 2021	47	14	16	16	31	14
March–June 2021	64	20	26	26	38	17
Maternal Age (years), mean (SD)	32.8 (7.3)		33.2 (6.2)		32.6 (7.8)	
Maternal Age (years)						
<30	57	18	15	15	42	19
30–<35	141	44	49	50	92	41
35 or older	122	38	34	35	88	40
Race and Ethnicity						
Non-Hispanic Black	24	8	6	6	18	8
Non-Hispanic Asian	34	11	14	14	20	9
Hispanic	30	9	6	6	24	11
Non-Hispanic White	203	63	62	63	141	64
Other ^2^	29	9	10	10	19	8
Education						
<College	41	13	12	12	29	13
College	95	30	24	24	71	32
Graduate/Professional sSchool	184	57	62	63	122	55
Essential Worker						
No	203	63	60	61	143	64
Yes	117	37	38	39	79	36

^1^ There were no statistically significant differences (*p* > 0.05) in the distributions of characteristics by pregnancy status (assessed by *t*-test for continuous variables or chi-square tests for categorical variables). ^2^ Other Race and Ethnicity includes women who identified as Native American/Alaska Native, Native Hawaiian/Pacific Islander, Other, or multiple races.

**Table 2 ijerph-19-05645-t002:** Mean (standard deviation, SD) number of types of personal care and household cleaning products and multivariable ^1^ linear regression analyses estimating adjusted mean differences in number of types of products used during the previous month by selected characteristics among women in the Maternal Health and Behavior Study (n = 320).

Characteristic	Personal Care Products				Household Cleaning Products			
	Mean	SD	Adjusted B	95% CI	Mean	SD	Adjusted B	95% CI
Time of Survey Completion								
July–October 2020	14.4	3.6	Reference		7.1	3.7	Reference	
November 2020–February 2021	15.1	3.1	−0.06	−1.26, 1.14	8.2	3.8	0.51	−0.69, 1.70
March–June 2021	15.2	3.3	0.61	−0.43, 1.65	7.6	3.3	0.16	−0.88, 1.20
Maternal Status								
Pregnant	14.8	3.3	0.08	−0.77, 0.93	7.0	3.4	−0.44	−1.29, 0.41
New Mom	14.7	3.6	Reference		7.5	3.7	Reference	
Maternal Age (years)								
<30	14.2	3.8	−0.79	−1.91, 0.33	7.9	4.1	0.03	−1.08, 1.15
30–<35	15.1	3.6	Reference		7.5	3.7	Reference	
35 and older	14.4	3.2	−0.64	−1.50, 0.22	6.9	3.2	−0.56	−1.41, 0.30
Race and Ethnicity								
Non-Hispanic Black	16.1	1.8	1.68	0.18, 3.18 ^2^	10.0	3.7 ^3^	2.74	1.24, 4.24 ^2^
Non-Hispanic Asian	15.1	3.8	0.80	−0.49, 2.08	7.1	3.9	0.20	−1.08, 1.49
Hispanic	15.2	4.0	1.19	−0.26, 2.64	7.7	3.8	0.26	−1.19, 1.71
Non-Hispanic White	14.3	3.5	Reference		6.9	3.4	Reference	
Other ^4^	15.3	3.2	1.25	−0.13, 2.63	8.2	3.5	1.22	−0.16, 2.60
Education								
<College	14.3	4.1	−0.80	−2.05, 0.45	8.1	3.5	0.60	−0.64, 1.85
College	15.1	3.7	0.58	−0.31, 1.47	7.8	3.6	0.92	0.03, 1.81 ^2^
Graduate/Professional School	14.6	3.2	Reference		7.0	3.6	Reference	
Essential Worker								
No	14.5	3.5	Reference		7.0	3.6 ^3^	Reference	
Yes	15.0	3.5	0.45	−0.40, 1.30	8.0	3.6	1.19	0.34, 2.04 ^2^

^1^ Models for personal care products and for household cleaning products are adjusted for all characteristics presented in the table. ^2^ *p* < 0.05, for B coefficient from multivariable linear regression model. ^3^ *p* < 0.05, F statistic from one-way ANOVA. ^4^ Other race and ethnicity includes women who identified as Native American/Alaska Native, Native Hawaiian/Pacific Islander, Other, or multiple races.

**Table 3 ijerph-19-05645-t003:** Bivariate analyses of change in use of antibacterial and organic personal care products (PCP) and household cleaning products (HCP) during the pandemic, receipt of healthcare provider advice on use of products, and opinions on environmental chemical exposures by selected maternal characteristics among women in the Maternal Health and Behavior Study (n = 320).

	Use Antibacterial PCP More Often ^1^	Use Antibacterial HCP More Often ^2^	Use Organic PCP More Often ^3^	Use Organic HCP More Often ^4^	Received PCP Advice ^5^	Received HCP Advice ^6^	Agree: Chemicals Pose Health Risks ^7^	Agree: Chemicals Impossible to Avoid ^8^
	n (%)	n (%)	n (%)	n (%)	n (%)	n (%)	n (%)	n (%)
All Women	107 (33)	171 (53)	73 (23)	83 (26)	74 (23)	52 (16)	302 (94)	257 (80)
Time of Survey Completion								
July–October 2020	70 (32)	118 (55)	44 (21)	52 (24)	44 (21)	29 (14)	202 (94)	171 (80)
November 2020–February 2021	18 (40)	19 (44)	11 (26)	10 (23)	12 (28)	11 (26)	41 (95)	36 (84)
March–June 2021	22 (34)	34 (53)	18 (28)	21 (33)	18 (28)	12 (19)	59 (92)	50 (78)
Maternal Status								
Pregnant	32 (33)	45 (46)	19 (19)	24 (24)	25 (26)	12 (12)	94 (96)	81 (83)
New Mom	75 (34)	126 (57)	54 (24)	59 (27)	49 (22)	40 (18)	208 (94)	176 (79)
Maternal Age (years)								
<30	17 (30)	27 (47)	16 (28) ^9^	16 (28)	14 (25)	12 (21)	53 (93)	46 (81)
30–<35	45 (32)	78 (55)	40 (28)	41 (29)	36 (26)	24 (17)	132 (94)	113 (80)
35 and older	45 (37)	66 (54)	17 (14)	26 (21)	24 (20)	16 (13)	117 (96)	98 (80)
Race and Ethnicity								
Non-Hispanic Black	9 (38)	16 (67) ^1^	8 (33)	9 (38)	7 (29)	6 (25) ^9^	24 (100)	22 (92)
Non-Hispanic Asian	9 (26)	19 (56)	11 (32)	12 (35)	6 (18)	7 (21)	34 (100)	29 (85)
Hispanic	15 (50)	22 (73)	10 (33)	9 (30)	11 (37)	10 (33)	28 (93)	22 (73)
Non-Hispanic White	60 (30)	94 (46)	37 (18)	45 (22)	42 (21)	24 (12)	188 (93)	161 (79)
Other ^10^	14 (48)	20 (69)	7 (24)	8 (28)	8 (28)	5 (17)	28 (97)	23 (79)
Education								
<College	11 (27)	21 (51)	12 (29)	13 (32)	12 (29)	11 (27)	38 (93)	33 (80)
College	33 (35)	57 (60)	26 (27)	25 (26)	21 (22)	18 (19)	87 (92)	74 (78)
Graduate/Professional School	63 (34)	93 (51)	35 (19)	45 (24)	41 (22)	23 (13)	177 (96)	150 (82)
Essential Worker								
No	67 (33)	109 (54)	43 (21)	50 (25)	51 (25)	30 (15)	191 (94)	162 (80)
Yes	40 (34)	62 (53)	30 (26)	33 (28)	23 (20)	22 (19)	111 (95)	95 (81)

^1^ Model compares use of antibacterial personal care products more often with reference, about the same/less often/never use. ^2^ Model compares use of antibacterial householding cleaning products more often with reference, about the same/less often/never use. ^3^ Model compares use of organic personal care products more often with reference, about the same/less often/never use. ^4^ Model compares use of organic householding cleaning products more often with reference, about the same/less often/never use. ^5^ Model compares receipt of advice regarding personal care product use from a healthcare provider with reference, no advice. ^6^ Model compares receipt of advice regarding household cleaning product use from a healthcare provider with reference, no advice. ^7^ Model compares agreement with statement that environmental chemicals pose health risks with reference, neither agree nor disagree/disagree/strongly disagree. ^8^ Model compares agreement with statement that environmental chemicals are impossible to avoid with reference, neither agree nor disagree/disagree/strongly disagree. ^9^ *p* < 0.05, chi square test. ^10^ Other race and ethnicity includes women who identified as Native American/Alaska Native, Native Hawaiian/Pacific Islander, Other, or multiple races.

**Table 4 ijerph-19-05645-t004:** Multivariable robust Poisson regression analyses ^1^ estimating adjusted prevalence ratios (aPR) for change in use of antibacterial and organic personal care products (PCP) and household cleaning products (HCP) during the pandemic (compared to before March 2020), receipt of healthcare provider advice on use of PCP and HCP, and agreement that environmental chemicals are impossible to avoid by selected maternal characteristics among women in the Maternal Health and Behavior Study (n = 320).

Characteristic	Use Antibacterial PCP More Often ^2^		Use Organic PCP More Often ^3^		Use Antibacterial HCP More Often ^4^		Use Organic HCP More Often ^5^		Received PCP Advice ^6^		Received HCP Advice ^7^		Agree: Chemicals are Impossible to Avoid ^8^	
	aPR	95% CI	aPR	95% CI	aPR	95% CI	aPR	95% CI	aPR	95% CI	aPR	95% CI	aPR	95% CI
Time of Survey Completion														
July–October 2020	1.00	Reference	1.00	Reference	1.00	Reference	1.00	Reference	1.00	Reference	1.00	Reference	1.00	Reference
November 2020–February 2021	0.93	0.58, 1.47	1.07	0.60, 1.89	0.66	0.46, 0.94	0.85	0.47, 1.54	1.16	0.64, 2.11	1.48	0.77, 2.83	1.05	0.91, 1.21
March–June 2021	1.07	0.71, 1.61	1.41	0.89, 2.24	1.00	0.76, 1.31	1.36	0.88, 2.09	1.62	0.98, 2.66	1.37	0.72, 2.60	0.95	0.82, 1.11
Maternal Status														
Pregnant	1.02	0.72, 1.42	0.74	0.47, 1.15	0.85	0.66, 1.08	0.88	0.58, 1.33	1.13	0.75, 1.71	0.66	0.37, 1.19	1.04	0.93, 1.16
New Mom	1.00	Reference	1.00	Reference	1.00	Reference	1.00	Reference	1.00	Reference	1.00	Reference	1.00	Reference
Maternal Age (years)														
<30	0.96	0.60, 1.55	0.88	0.53, 1.44	0.84	0.62, 1.15	0.91	0.55, 1.50	0.92	0.53, 1.61	1.00	0.54, 1.87	1.00	0.86, 1.17
30–34	1.00	Reference	1.00	Reference	1.00	Reference	1.00	Reference	1.00	Reference	1.00	Reference	1.00	Reference
35 and older	1.19	0.85, 1.67	0.47	0.28, 0.79 ^10^	0.99	0.80, 1.23	0.72	0.47, 1.11	0.77	0.48, 1.21	0.76	0.42, 1.35	1.00	0.88, 1.13
Race and Ethnicity														
Non-Hispanic Black	1.36	0.79, 2.35	1.70	0.96, 2.99	1.57	1.15, 2.13 ^10^	1.70	0.98, 2.96	1.38	0.68, 2.79	1.78	0.85, 3.75	1.15	0.99, 1.33
Non-Hispanic Asian	0.88	0.48, 1.63	1.96	1.13, 3.40 ^10^	1.26	0.90, 1.77	1.67	0.99, 2.82	0.86	0.40, 1.87	1.79	0.83, 3.85	1.07	0.91, 1.25
Hispanic	1.97	1.22, 3.19 ^10^	1.59	0.90, 2.80	1.83	1.35, 2.47 ^10^	1.34	0.73, 2.46	1.79	0.97, 3.31	2.07	1.02, 4.19 ^10^	0.91	0.73, 1.13
Non-Hispanic White	1.00	Reference	1.00	Reference	1.00	Reference	1.00	Reference	1.00	Reference	1.00	Reference	1.00	Reference
Other ^9^	1.73	1.11, 2.68 ^10^	1.33	0.65, 2.74	1.64	1.25, 2.15 ^10^	1.27	0.66, 2.43	1.31	0.71, 2.40	1.37	0.58, 3.25	0.99	0.81, 1.21
Education														
<College	0.67	0.39, 1.17	1.23	0.69, 2.20	0.91	0.63, 1.30	1.16	0.67, 2.01	1.05	0.59, 1.88	1.58	0.77, 3.24	1.00	0.85, 1.18
College	1.01	0.72, 1.43	1.41	0.90, 2.20	1.19	0.96, 1.48	1.07	0.70, 1.66	0.91	0.57, 1.46	1.41	0.81, 2.44	0.96	0.84, 1.09
Graduate/Professional School	1.00	Reference	1.00	Reference	1.00	Reference	1.00	Reference	1.00	Reference	1.00	Reference	1.00	Reference
Essential Worker														
No	1.00	Reference	1.00	Reference	1.00	Reference	1.00	Reference	1.00	Reference	1.00	Reference	1.00	Reference
Yes	1.02	0.73, 1.43	1.05	0.69, 1.61	0.99	0.80, 1.22	1.01	0.68, 1.51	0.66	0.41, 1.06	1.17	0.68, 1.98	1.03	0.92, 1.15

^1^ Each logistic regression model includes all of the characteristics listed in the table. ^2^ Model compares use of antibacterial personal care products more often with reference, about the same/less often/never use. ^3^ Model compares use of antibacterial householding cleaning products more often with reference, about the same/less often/never use. ^4^ Model compares use of organic personal care products more often with reference, about the same/less often/never use. ^5^ Model compares use of organic householding cleaning products more often with reference, about the same/less often/never use. ^6^ Model compares receipt of advice regarding personal care product use from a healthcare provider with reference, no advice. ^7^ Model compares receipt of advice regarding household cleaning product use from a healthcare provider with reference, no advice. ^8^ Model compares agreement with statement that environmental chemicals are impossible to avoid with reference, neither agree nor disagree/disagree/strongly disagree. ^9^ *p* < 0.05. ^10^ Other race and ethnicity includes women who identified as Native American/Alaska Native, Native Hawaiian/Pacific Islander, Other, or multiple races.

## Data Availability

Data may be available upon request and appropriate approvals from study investigators and appropriate IRB.

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
