# Peer review of "Personal Care and Household Cleaning Product Use among Pregnant Women and New Mothers during the COVID-19 Pandemic"

_ijerph, 2022, doi:10.3390/ijerph19095645_

Round 1

Reviewer 1 Report

This is a very interesting observational study regarding the personal care and household cleaning product use among pregnant women and new mothers in New York City during the COVID-19 pandemic. The authors also queried women’s change in antibacterial product use, opinions on environmental chemicals, and receipt of healthcare provider advice regarding exposures during this time. The results show that women increased their use of antibacterial products in response to the pandemic. Differences were observed by women’s race/ethnicity, level of education, and employment as an essential worker. From my personal opinion, the manuscript can be accepted for publication after addressing the following questions:

  1. Please briefly describe the definition of “essential workers”. Because job functions may play roles for the hygiene product use.
  2. In Abstract, line 19, it’s confusing for the description of “Non-Hispanic Black, Hispanic, and women with other race/ethnicity…”. Does it mean all the women?

Author Response

  1. Please briefly describe the definition of “essential workers”. Because job functions may play roles for the hygiene product use.
  • We now provide our exact survey question in the text. We did not collect information on the specific job type, only whether they were considered essential workers during the pandemic.

  1. In Abstract, line 19, it’s confusing for the description of “Non-Hispanic Black, Hispanic, and women with other race/ethnicity…”. Does it mean all the women?
  • We clarified this issue throughout the text. Other refers to women who reported their race and ethnicity as Native American/Alaska Native, Native Hawaiian/Pacific Islander, “Other”, and women who chose more than one race.

Author Response

Introduction:

Lines 54-60: The objectives of the study state associations will be evaluated, but for what? This is not clearly stated. Evaluated for increased use? Change in use patterns during various intervals (e.g., prior to vs. during pandemic, etc.). Evaluated for differences among race/ethnicity? Healthcare provider advice is a very interesting concept to “evaluate” in this manuscript, but further detail could be provided regarding how the authors met this objective.

  • It is not clear what the reviewer’s comment refers to in the text – we did not use the term “evaluate”. We state our objectives in the introduction section as the following: “In the current study, our objective was to examine personal care and household cleaning product use among pregnant women and new mothers in New York City during the first year of the pandemic. We also examined women’s attitudes towards environmental chemicals and their receipt of healthcare provider advice regarding product use during this time.” We did not have specific a priori hypotheses about product use.

Materials & Methods:

Lines 83-85: Consider the addition of how products were selected, chemical constituents, and

definitions. For example, the Introduction seems to suggest some personal care products are cosmetic and would decrease during the pandemic. Would it be worthwhile to group cosmetic products (containing parabens) separately from soap or personal hygiene products? How were the products chosen for the questionnaire?

  • In the introduction, we mention the population-level data regarding purchasing behaviors during the pandemic as general information for the reader– we couldn’t find any studies on individual level changes during this time. Our survey queried health behaviors, such as physical activity and dietary intakes, during the pandemic, as well as product use. We wanted to ask about both personal care products and household cleaning products, while being aware of time constraints for participants. We consulted several sources regarding the list of products that should be included. We decided to use products included in the Early Life Exposures Assessment Tool, as well as those from other studies (now referenced) – creating a final list of 22 types of personal care products and 15 types of household cleaning products. Previous studies do not group products by possible chemical constituents, unless there is some a priori objective to compare product use with specific measured chemical biomarkers. Given that we queried broad types of products and lack specific brand information, chemical constituent analyses, and biomarker data, our analyses focus on number of types of products used. We chose not to ask about specific cosmetics (as done by Dodd et al. which focused only on personal care products) to limit the time for survey completion; for example, we only asked about “face makeup” rather than individual categories like eye liner, shadows, concealers, etc.
  • We now provide the following information in the text:

“A list of 22 types of personal care products and 17 types of household cleaning products was adapted from the Early Life Exposures Assessment Tool and previous studies. Personal care products included soaps, body and face lotions, hair styling products, deodorant, sunscreen, nail polish, and perfume (full list in Supplemental Figure 1a). Household cleaning products included bleach, ammonia, multi-use cleaners (without bleach or ammonia), other cleaning sprays, and air fresheners (full list in Supplemental Figure 1b). Women reported their frequency of use (daily, weekly, monthly, never, or don’t know) of products during the previous month.”

Lines 88-89: Was the term, “environmental chemical” defined on the questionnaire? Were “health risks” defined as relevant to pregnancy/early development, etc. ? How did the authors assure quality control over the questionnaire responses? Do the authors feel as though endocrine-disrupting chemicals were the human health risk evaluated in the questionnaire? If so, was this communicated via the questionnaire?

  • We now provide the exact statements that were included in the survey - we refer to them as “chemicals in the environment”. We did not ask specifically about endocrine-disrupting chemicals (which are not the only types of chemicals that we are exposed to in the environment) and we did not provide a definition of environmental chemicals. These questions are very similar to those used in the TIDES study (Barrett et al.). We now provide the reference.
  • “Lastly, they were asked about their opinions of environmental chemicals by their level of agreement (strongly agree, agree, neither, disagree, strongly disagree) with the following statements: “Chemicals in the environment can pose health risks.” and “Chemicals in the environment are in so many things that it’s impossible to avoid them.” These questions were adapted from those used in The Infant Development and the Environment Study (TIDES).”

Line 102: Seasonality should be further defined. It is unclear how these categorizations are used in the analysis of data, or how these intervals differ based on vaccine availability. How is July-Oct 2020 different from Mar-Jun2021? If dates are used as predictors in the models described below, what are your predictions? The reader cannot discern which interval has greater vaccine rates, for instance.

  • We categorized survey administration to broadly reflect three factors related to the pandemic: 1) seasonality - July – October 2020 and March – June 2021 are warmer months, summer/fall and spring/summer, respectively, compared to the colder months of November 2020 – February 2021, when COVID cases may be higher due to greater indoor activities; 2) vaccines became broadly available in NYC beginning in March 2021, which we have included parenthetically in the text; and 3) regional infection rates in New York City. We didn’t have a priorihypotheses for each of the characteristics examined in the analyses and how they would influence product use – but we thought that timing of survey administration in relation to the progression of the COVID-19 pandemic was important to consider.
  • “The time period of survey administration was categorized as July–October 2020, November 2020–February 2021, and March–June 2021, to generally reflect three pandemic-related factors: seasonality (changes in weather and temperature to reflect spring/summer/fall versus winter, when concern about COVID transmission may be higher); vaccine availability (vaccines became broadly available beginning in March 2021); and regional infection rates.”

Line 107: Greater detail is needed regarding the statistical analyses. It is not clear what variables were included in the pairwise comparisons (t-test), groupwise comparisons (ANOVA), or how the associations (linear regression) were defined or expressed. Are correlation coefficients available? What were the dependent, independent variables?

  • We clarified our methods section and the information provided in the text to address this comment and the following methods-related comments.
  • “All statistical analyses were conducted Stata/SE 15.1 (StataCorp LP, College Station, TX, USA). We performed descriptive analyses of women’s characteristics for the entire sample and stratified by maternal status, currently pregnant or new mother, using t-tests (continuous variables) or chi-square tests (categorical variables). We used one-way ANOVA tests to examine differences in the number of types of personal care products and the number of types of household cleaning products used during the previous month (means and standard deviations, SD, of number of types of products used) by the selected characteristics (timing of survey administration, maternal status, age, race and ethnicity, education, and essential worker status). Multivariable linear regression models estimated adjusted mean differences in the number of types of personal care products and household cleaning products used (separate models) with the selected characteristics (models included all characteristics simultaneously).

Chi-square tests examined bivariate associations of women’s change in use of antibacterial and organic products during the pandemic (more often compared to about the same/less often/never, reference), receipt of healthcare provider advice (yes/somewhat compared to no, reference), and opinions of environmental chemicals (strongly agree/agree compared to neither/disagree/strongly disagree, reference) with the selected characteristics. We used separate multivariable robust Poisson regression models to estimate adjusted prevalence ratios (aPR) for change in product use (antibacterial and organic personal care and household cleaning products, respectively), receipt of healthcare provider advice, and agreement that environmental chemicals are impossible to avoid with the selected characteristics (models included all characteristics simultaneously). We did not examine associations for agreement that environmental chemicals can pose health risks, since 94% of women reported agreement.”

Line 110: MVLIN regression model: predictor variables are time period, education, essential worker, etc.? Please clarify which variables are included in each model. Separate models for PPC? HCP? And opinion? Or are all included in the model? Please clarify methods.

  • This has been clarified in the methods and tables.

Line 113: MVLOG regression model: See previous comment. What variables were included in each model?

  • This has been clarified in the methods and tables.

Line 114: Were separate models used for each category? PCP, and HCP? This is not clear.

  • Separate models were used. This has been clarified.

Line 115: Please provide version of R used in the analyses.

  • We now provide the correct statistical software, since we redid all of the models.

Results:

Table 1: Would the study design benefit from a control group to further differentiate the associations (e.g., non-pregnant women of reproductive age)?

- Unfortunately, we do not have a control group for this study. We took advantage of data that was being collected by the COPE study, a cohort of pregnant women and new mothers established during the COVID-19 pandemic. Data was not collected from women who were not pregnant during this time period.

Line 125: It is not clear what comparisons were made; what statistical tests were used. Also, please correct typographical error (p<0.05).

  • That is not a typo. The distributions of characteristics did not vary by maternal status. This is now clarified in the footer: “There were no statistically significant differences (p>0.05) in the distributions of characteristics by maternal status (assessed by t-test for continuous variables or chi-square tests for categorical variables).”

Figure 1a. Are the constituents (chemicals of concern) in any of the PCP's presented in relation to potential risk? Consider greater interpretation of trends noted in this graphic. Is there significance in regard to non-antibacterial soaps, deodorants being the most commonly used? (based on risk, chemical constituents, etc.?) Trends in this graphic are not well characterized in the text. Are there different risks associated with different exposures (dermal vs inhalation)?

  • We do not have any information on trends in use of products. This graphic was intended to be part of Supplemental information and used to display the types of products that were included in the survey, as well as women’s reported frequency of use of these products during the previous month. We did not collect any specific information on brands and cannot make any assumptions regarding their chemical profiles.

Lines 126-127: Consider defining “multi-use cleaner” and potential ingredients.

  • We clarified this in the text as it was stated in the survey: “multi-use cleaners that do not contain bleach or ammonia”

Line 148: Was more than one Lin Regression model fit to the data presented in Table 2?

  • This is now clarified in the methods and the table footer. We used separate linear regression models for personal care products and for household cleaning products. In each of the models, the dependent variable is the number of products used and the independent variables are the selected characteristics (timing of survey completion, maternal status, age, race and ethnicity, education, essential worker).

Line 152: please clarify “all variables”

  • This has been clarified.

Line 157: “During the pandemic vs Before the pandemic”: Are these intervals defined previously? Clarification in parentheses may be helpful to the reader.

  • This is stated in the methods – “before the COVID-19 pandemic (March 2020) to the current time period at survey administration”. We added “(March 2020) to this sentence for the reader’s convenience.

Line 160: Please define aOR.

  • This is has been changed in the text.

Table 2: Please define “Reference” as used in table, statistical comparisons, etc.. Why/how were these categories selected as reference groups? (same comment for other Reference groups). Additional clarification may be warranted in the M&M’s section.

  • This information was added to the methods section and to the footer.

Table 2, Essential Worker, 6.9   3.53 (SD of HCP’s): Should superscript 3 be on the non-Reference value?

  • This was fixed.

Table 3: Bivariate analyses are not clarified. Additional detail is needed regarding which variables were included in the various models, or model.

Table 4: Multivariable logistic regression analyses estimating odds of change in use of antibacterial and organic personal care products (PCP) and household cleaning products (HCP) during the pandemic… Are all 3 variables included in a single model? (antibacterial PCP, Organic PCP, HCP). Identify which products are considered antibacterial, organic, etc.

  • This has been clarified in the text.

Table 4 Title: “…during the pandemic”: is this interval defined in the table? Is this the reference

group? Please provide clarification.

  • This has been clarified in the title.

Table 4, footnote a: Please consider providing further clarification for “not in agreement. “ This

statement is not clear, as written.

  • This has been clarified.

Discussion:

The authors should consider providing greater discussion as to if/how the objectives explicitly stated in the Introduction were met. Revisit organization of those ideas in relation to highlighting key issues. Were the associations “evaluated” in the study clearly characterized? (what was the association with pregnancy status, for example)

  • We redid the analyses to include the results considering pregnancy status. There were no differences observed for the selected outcomes by pregnancy status.

Lines 213-214: not all products listed are associated with endocrine-disrupting chemicals. This list contains a broad range of chemical constituents, likely with vastly different risk factors.

  • In these lines we don’t state that these products contain endocrine-disrupting chemicals. We state only that the cumulative results from studies suggest that “women use a range of chemical-containing products on a daily basis”. In the following paragraph, we state that product use has been associated with chemical body burden of specific chemicals.

Line 221-222: This is a speculative statement. Please clarify why the replacement chemicals are

considered “potentially harmful.” Do they contain similar chemical properties to triclosan?

  • We provide a reference for this statement that replacement chemicals for triclosan may also have similar harmful effects as triclosan. We edited this statement based on the following comment.

Line 224: What is the endrocrine-related chemical constituent of concern in antibacterial products? Define the human health concern/risk associated w/ antibacterials in relation to use (did it increase?) during the pandemic?

  • We aren’t stating that all of these products contain chemicals with endocrine-disrupting properties (but it should be noted that this term covers an extremely broad range of health effects). We edited our text to clarify that the results of our study showed that a percentage of women increased their use of products labeled as anti-bacterial or organic (the specific survey questions are now provided in the methods) and that these products may contain fragrances and other chemicals. We referenced an article regarding triclosan replacement chemicals that states these replacement chemicals may also have harmful health effects. We also edited this text in response to this comment.

Lines 229-244: Good use of references in support of these statements.

  • Thank you.

Lines 247-250: Important points to consider. If possible, the authors should consider follow-up

studies regarding this topic.

  • We are not considering a follow-up study at the moment.

Line 256: Consider adding “when” to minimize exposures since your study included temporal factors (pregnancy, new moms, not pregnant, pre-pandemic, during pandemic). Hazard vs risk trade-offs to product use vs viral protection?

  • Our study focused on a population of pregnant women and new mothers, who represent an at-risk population for environmental chemical exposures that has been studied in the literature. There is evidence that environmental chemical exposures likely influence the health of individuals throughout the lifespan – infancy, childhood, puberty, reproductive years, menopause (for females) – and the importance of lessening exposures should not only be emphasized among pregnant women (or even women in general). We are unaware of any studies that have evaluated the trade-offs between product use versus viral protection and we do not feel comfortable speculating on this issue. In our population, we found that some women reported using products more often, they expressed concerns about environmental chemical exposures, and they were not having discussions regarding product use/environmental chemicals with their healthcare providers. We focused our recommendations in the discussion on healthcare providers and available resources that may be disseminated to this patient population.

The authors might consider elaborating on trends associated with certain product types (increased anti-microbial product use during the pandemic.; decreased PCP use such as cosmetics during the pandemic).

  • We only asked about change in use of products labeled as antibacterial and organic – we have no information on changes in use of specific types of products during this time.

Line 273: Specific chemical constituent classes may also have been helpful.

  • This has been added to the text.

Lines 278-279: Authors should consider whether testing was available during all intervals evaluated in this study. Two-thirds of women responded when cases were low, but testing may not have been available at that time.

  • Testing was available in New York City during July – October 2020, although we do not know how many women were actively getting tested. We changed the text to specify that these women were responding during the summer and fall months - we are acknowledging that case counts were low during this time based on available city data.

Reviewer 3 Report

Abstract is not written in full sentences and it is very confusing, thus it needs major rewriting. It is unaccaptible to use terms such as „women of color“.

It is important to report how many women were approached and how many accepted to take part in study.

I wonder whether authors read own manuscript. The first sentence in chapter Discussion

„Authors should Among 328 pregnant women and new moms in New York City during the COVID-19 pandemic, women on average reported using approximately 15 types of personal care products (out of 22 types) and 7 types of household products (out of 17 types) during the previous month.“ ?????

What does it mean „non-Hispanic Black women“?

It is important to include ethnical differences in sex hormone levels and aromatase levels what is crucial in biological effects and associated ethnically specific health risks.

Manuscript needs significant improvement in using uniform terms for definition of different ethnic groups which took part in the study.

Author Response

Abstract is not written in full sentences and it is very confusing, thus it needs major rewriting. It is unaccaptible to use terms such as „women of color“.

  • The abstract is written in complete sentences, but we did cut some information in order to meet the word limit. We edited it to make it less confusing. We also removed the term “women of color” since we agree it is outdated.

It is important to report how many women were approached and how many accepted to take part in study.

  • We added this information. There were 424 women who completed the second assessment of COPE and 328 women (77%) completed the MHBS survey.

I wonder whether authors read own manuscript. The first sentence in chapter Discussion

„Authors should Among 328 pregnant women and new moms in New York City during the COVID-19 pandemic, women on average reported using approximately 15 types of personal care products (out of 22 types) and 7 types of household products (out of 17 types) during the previous month.“ ?????

  • We did read our own manuscript – that was left over from the IJERPH template and was simply a mistake. It was been deleted.

What does it mean „non-Hispanic Black women“?

  • The survey used race and ethnicity questions as specified by the NIH. Non-Hispanic Black women are those who reported their race as black and their ethnicity as non-Hispanic.

It is important to include ethnical differences in sex hormone levels and aromatase levels what is crucial in biological effects and associated ethnically specific health risks.

  • Our study was a survey regarding women’s use of household cleaning and personal care products (which may be sources of exposures to endocrine disrupting chemicals); changes in antibacterial product use; opinions on environmental chemicals; and receipt of healthcare provider advice. We did not measure biomarkers of chemicals or levels of aromatase or sex hormones in these women (it should be noted that aromatase and sex hormone levels are influenced by numerous factors other than race and ethnicity, such as age, body fat, underlying medical conditions). A discussion of possible sex hormone levels or aromatase levels, particularly by race and ethnicity, is beyond the scope of this study and there is not strong evidence in the literature on this topic.

Manuscript needs significant improvement in using uniform terms for definition of different ethnic groups which took part in the study.

  • We clarified how race and ethnic groups were reported by women on the survey. Women reported their race and ethnicity as non-Hispanic Black, non-Hispanic Asian, non-Hispanic
    White, Hispanic, Native American/Alaska Native, Native Hawaiian/Pacific Islander, or “Other”. Women could select more than one race. For analyses, we categorized these as non-Hispanic Black, non-Hispanic Asian, non-Hispanic White, Hispanic, and Other race. Other race included Native American/Alaska Native, Native Hawaiian/Pacific Islander, “Other”, and women who chose more than one race.

Round 2

Reviewer 3 Report

I am satisfied with all improvement except I have to insist in adding of ethnically specific response on endocrine disruptors as this fact is critical in evaluation of health risk. Please use all or some of suggested references

https://pubmed.ncbi.nlm.nih.gov/24285681/

https://academic.oup.com/jcem/article/96/10/3199/2834917

https://www.nature.com/articles/6691082.pdf?origin=ppub

https://pubmed.ncbi.nlm.nih.gov/16949385/

https://pubmed.ncbi.nlm.nih.gov/19240151/

https://pubmed.ncbi.nlm.nih.gov/34508608/

Author Response

We understand the reviewer’s comment regarding potentially different responses to environmental endocrine disruptors by women’s race and ethnicity. However, none of the provided studies specifically examined ethnic differences in exogenous EDC exposures (from products or other sources) and subsequent changes in hormonal activity in women. Instead they focus on differences in endogenous levels (e.g., gene expression, enzyme activity, androgen and estrogen concentrations) - they also have varying methodology and study populations (such as post-menopausal women). It is not clear how the findings of the provided studies contribute to the discussion of product use in the current study. Racial and ethnic differences in product use (including the types of products used, their frequency and duration of use, and their chemical constituents) have been proposed as plausible risk factors for health disparities and explain part of the increased risk of hormonally-mediated diseases, particularly among Black women. Dr. Tamarra James-Todd and colleagues have written about this issue extensively in the literature and we had mentioned it in the Discussion section. In response to your comment, we edited the text as follows:  

“Racial and ethnic differences in product use (such as types, frequency of use, and duration), as well as endogenous hormonal activity, have been documented and may contribute to disparities in risk of hormonally-mediated diseases [15,22,31,32]. For example, greater personal care product use, especially hair products, and higher exposures to chemicals from personal care products are often observed among Black women and are linked to increased risk of breast cancer, and other reproductive health-related conditions, in this population [2,15,22,33-35].”